# Quantitative Assessment of the Trigger Effect of Proton Flux on Seismicity

**DOI:** 10.3390/e27050505

**Published:** 2025-05-08

**Authors:** Alexey Lyubushin, Eugeny Rodionov

**Affiliations:** Institute of Physics of the Earth RAS, Moscow 123242, Russia; evgeny_980@list.ru

**Keywords:** earthquakes, proton flux, point processes, wavelets, kurtosis, spectral slope, entropy, multifractals

## Abstract

An estimate of the trigger effect of the proton flux on seismicity was obtained. The proton flux time series with a time step of 5 min, 2000–2024, was analyzed. In each time interval of 5 days, statistics of the proton flux time series were calculated: mean values, logarithm of kurtosis, spectral slope, singularities spectrum support width, wavelet-based entropy, and the Donoho–Johnston wavelet-based index. For each of the used statistics, time points of local extrema were found, and for each pair of time sequences of proton flux statistics and earthquakes with a magnitude of at least 6.5 in sliding time windows, the “advance measures” of each time sequence relative to the other were estimated using a model of the intensity of interacting point processes. The difference between the “direct” measure of the advance of time points of local extrema of proton flux statistics relative to the time moments of earthquakes and the “inverse” measure of the advance was calculated. The maximum proportion of the intensity of seismic events for which the proton flux was a trigger was estimated as 0.28 for using the points of the local minima of the singularities spectrum support width.

## 1. Introduction

The influence of solar activity on various processes on Earth has long been the subject of close study, which resulted in the appearance of the term “space weather”. The review [1] considers various aspects of the influence of solar activity on the Earth’s climate and anthropogenic processes. The ionosphere, as an important component of the concept of space weather, was studied in the works [2,3,4]. The results of the development of statistical methods for predicting strong solar flares, including using machine learning, are presented in the articles [5,6,7,8]. An important issue in the study of space weather is its impact on catastrophic events in the life of society, such as earthquakes. In particular, methods have not yet been developed that allow us to unambiguously answer the question of whether strong solar flares and other electromagnetic events in the ionosphere have a trigger effect on the occurrence of sufficiently strong earthquakes. A lot of research is devoted to this issue [9]. The papers [10,11] present the results of the analysis of correlations between 11 and 12-year cycles of solar activity and time intervals of increasing intensity of seismic events over long periods of time. A comparison of time intervals of high seismic activity with the phases of solar cycles since 1900 is carried out in the paper [12].

The identification of the effects of the delay of strong earthquakes relative to the time intervals of geomagnetic storm maxima was considered in [13,14]. In [15], a similar effect of the delay of seismic events was studied for sunspot numbers. The hypothesis about the occurrence of time anomalies of atmospheric electric fields preceding the occurrence of strong earthquakes, including deep-focus ones, as a result of processes in the source of an impending seismic event was studied in [16]. A similar question about the occurrence of atmospheric and ionospheric electromagnetic signals recorded by spacecraft and preceding moderate seismic events was considered in [17]. In [18,19], the difference between the global seismic process and the Poisson one after excluding aftershocks is explained by the piezoelectric effect in rocks as a result of the impact of the proton flux during solar activity, which has a periodic time structure in addition to the 11–12-year solar cycle. In [20,21,22], the hypothesis is investigated about the generation of telluric currents in the Earth’s crust as a result of the impact of disturbances of ionospheric electromagnetic fields from solar flares and, as a consequence, their trigger effect on the foci of future seismic events. A statistical analysis of the impact of 50 largest solar flares in the time interval 1997–2024 on global seismic activity was performed in [23], as a result of which an increase in seismic activity was discovered within 10 days after the flare compared to 10 days before it. The paper [24] provides an overview of the work in Russia for the period 1995–2020 on the study of the influence of artificial and natural electromagnetic impacts on seismicity and discusses possible ways of using electromagnetic seismicity to reduce seismic hazard. The classification of seismic events with a magnitude of at least 6 as they occur as a result of the impact of a proton flux using a neural network was performed in the paper [25].

The complex dynamics of both the Sun and solar-terrestrial relations requires the use of a set of modern data processing methods based on the use of nonlinear models for the analysis of time series describing interacting systems [26]. In [27,28], the internal dynamics of solar cycles were studied using methods of empirical orthogonal oscillation modes, estimates of their maximum Lyapunov exponents, and entropy flows between the values of various parameters of processes inside the Sun. In [29,30], various estimates of the Hurst exponent and entropy measures were used to analyze data obtained using the Swarm satellite network of the European Space Agency to describe the most intense magnetic storms and to quantitatively study the complexity of processes in the upper ionosphere. In [31], a study was conducted of the structure of currents induced by geomagnetic storms, leading to accidents in electrical networks, by applying information theory and various entropy measures to their time series.

In this paper, a new method is proposed that allows obtaining a quantitative estimate of the influence of various statistics of the proton flux density time series measured by the Solar Heliospheric Observatory (SOHO) [32] on the sequence of earthquakes with a magnitude of at least 6.5. The method is based on the use of estimates of “advance measures” based on a parametric model of the intensities of interacting point processes and on the calculation of wavelet measures of spectral tilt and entropy, as well as on an estimate of the width of the carrier of the multifractal spectrum of singularities of the proton flux density time series.

## 2. Proton Flux Initial Data

The time series of proton flux values with a time step of 5 min was downloaded from the website [33]. Figure 1 shows a graph of the time series of proton flux density for the time period from the beginning of 2000 to 17 October 2024. The time step is 5 min.

In the future, when analyzing the relationship between the proton flux and earthquakes, we will use various statistics of the proton flux density time series calculated in a time interval of 5 days (1440 readings with a time step of 5 min), taken with an offset of 1 day (288 5-min readings).

We start with the simplest statistics equal to the average value of the flux density in these intervals. From the beginning of 2000 to 17 October 2024, 1136 earthquakes with a magnitude of at least 6.5 occurred. Therefore, we found the 1136 largest local maxima of the average flux density values and present them as a function of the position of the right end of 5-day time windows. These two time sequences of events are shown in Figure 2.

The chosen minimum magnitude of 6.5 is representative for the whole world, and the corresponding time sequence does not contain aftershocks, which is important to ensure homogeneity of the time points of seismic events.

## 3. Periodic Components of the Proton Flux

The proton flux comes from the Sun, for which a number of periodicities are known, the best known of which is the 11–12-year periodicity, which determines the numbering of solar cycles. However, there is also a period due to the rotation of the Sun around its axis, equal to approximately 27 days. This periodicity should be reflected in variations in the proton flux density. Figure 3 shows the spectral composition of the proton flux density after the transition from the original time series to the average daily values. The power spectrum of the proton flux was calculated in a sliding time window of 730 days (2 years) with an offset of 30 days using the autoregressive model of order 70 [35]. The time–frequency diagram of the evolution of the spectrum logarithm is shown in Figure 3a, while the values obtained by averaging the spectral estimates from all time windows are presented as a graph in Figure 3b. This figure highlights four maximum spectral peaks, next to which the values of their periods in days are indicated. It is evident that the maximum spectral peak had a period of 26.95 days, approximately equal to the period of the Sun’s rotation. The other periods correspond to its overtones.

## 4. Periodic Components of the Earthquake Sequence

Of interest is the question of whether the 27-day periodicity of the proton flux shown in Figure 2 is reflected in the periodicity of the intensity of seismic events. To do this, it is necessary to estimate this periodicity, taking into account that the sequence of earthquakes is not a time series with a constant time step, for which classical spectral estimation methods [35] can be applied. Below, the method proposed in [36] is used to estimate the periodic components of the intensity of the sequence of events. In [37], this method was used to calculate the periodic component of the stepwise variations in the time series of the displacement of the earth’s surface measured by GPS.

Let $t_{i},\;i=1,\ldots ,N$ be the times of the sequence of events observed on the interval (0,T]. Consider the following intensity model containing a periodic component:(1)λ(t)=μ⋅(1+acos(ωt+φ))
where frequency ω, amplitude a, 0≤ a ≤ 1, phase angle φ, and φ∈[0,2π] multiplier μ>0 (describing the Poisson part of the intensity) are parameters of the model. Thus, the Poisson part of the intensity is modulated by a harmonic oscillation. Let us fix some value of frequency ω. The logarithmic likelihood function [38] in this case for a series of observed events is equal to(2)$$\begin{array}{l} \ln(L)=\sum_{t_{i}}\ln(\lambda (t_{i}))-\int_{0}^{T}\lambda (s)ds=\\ =N\ln(\mu )+\sum_{t_{i}}\ln(1+a\cos(\omega t_{i}+\varphi ))-\mu T-\frac{\mu a}{\omega }[\sin(\omega T+\varphi )-\sin(\varphi )] \end{array}$$

Taking the maximum of expression (2) with respect to the parameter μ, it is easy to find that(3)$$\ln(L(\hat{\mu },a,\varphi |\omega ))=\sum_{t_{i}}\ln(1+a\cos(\omega t_{i}+\varphi ))+N\cdot \ln(\hat{\mu }(a,\varphi |\omega ))-N$$

It should be noted that the expression $\hat{\mu }(a=0,\varphi |\omega )\equiv {\hat{\mu }}_{0}=N/T$ is an estimate of the intensity of the process under the condition that it is Poisson homogeneous (purely random). Thus, the increment of the logarithmic likelihood function due to the consideration of a richer intensity model with a harmonic component with a given frequency ω than for a purely random flow of events is equal to(4)$$\Delta \ln L(a,\varphi |\omega )=\sum_{t_{i}}\ln(1+a\cos(\omega t_{i}+\varphi ))+N\cdot \ln(\hat{\mu }(a,\varphi |\omega )/{\hat{\mu }}_{0})$$

Let(5)$$R(\omega )=\underset{a,\varphi }{\max}\Delta \ln L(a,\varphi |\omega ),\;\;0\leq \;a\;\leq \;1,\;\varphi \in [0,2\pi ]$$

An important issue when applying this method to real data is determining the statistical significance of the obtained peak values of statistics (5). Let us consider two hypotheses for the same data set $X^{\left(N\right)}$ consisting of independent observations:

(1)$X^{\left(N\right)}$ distributed by density $p_{0}(X^{\left(N\right)}|\theta_{0})$—hypothesis $H_{0}$;(2)$X^{\left(N\right)}$ distributed by density $p_{1}(X^{\left(N\right)}|\theta_{1})$—hypothesis $H_{1}$.

Here, $\theta_{0}$ and $\theta_{1}$ are vectors of unknown parameters, having dimensions $m_{0}$ and $m_{1}$, and the hypothesis $H_{1}$ is more “rich”: $m_{1}>m_{0}$, and the vector of parameters $\theta_{1}$ completely include the components of the vector $\theta_{0}$. Let us consider the difference between the logarithms of the likelihood for these two hypotheses, provided that the vectors of parameters are taken from their maximum likelihood estimates:(6)$$\Delta \ln L(X^{\left(N\right)})=\ln\left(\underset{\theta_{1}}{\max}\;p_{1}(X^{\left(N\right)}|\theta_{1})\right)-\ln\left(\underset{\theta_{0}}{\max}\;p_{0}(X^{\left(N\right)}|\theta_{0})\right)$$

It is evident that $\Delta \ln L(X^{\left(N\right)})\geq 0$. According to Wilks’ theorem [39], if the hypothesis is true, the quantity (6) has an asymptotic distribution:(7)$$\Delta \ln L(X^{\left(N\right)})∼\frac{\chi_{m}^{2}}{2},\;m=m_{\;1}-m_{0},\;N\to \infty$$

In our case, m=2 and therefore, the doubled value (8) has an asymptotic distribution density $\chi_{2}^{2}$ equal to $e^{-x/2}/2$, and the value (8) itself is distributed asymptotically as(8)$$\mathrm{Prob}\{R(\omega )<x\}=1-e^{-x},\;N\to \infty$$
provided that the analyzed sequence of time moments is distributed according to the Poisson law with constant intensity. Expression (8) allows us to set thresholds for statistics that allow us to assert that only when they are exceeded does the sequence of time moments differ from the Poisson sequence with a given probability.

For a time sequence of seismic events with a magnitude of at least 6.5 (Figure 2b), we calculated the increments of the logarithmic likelihood function (5) in a sliding time window of 730 days (2 years) with a shift of 30 days for 200 frequency values ω corresponding to the values of periods varying from 10 to 100 days with a uniform step in a logarithmic scale. The resulting time–frequency dependence, similar to the usual spectral time–frequency diagram in Figure 3a, is shown in Figure 4a.

Figure 4b shows the graph of averaging the increments of the logarithmic likelihood function (4) for all time windows and highlights five “spectral” peaks exceeding level 2 with periods of 14.84, 23.89, 26.64, 50.5, and 74.9 days. For them, the peak values of the average increments ΔlnL were 2.08, 2.14, 2.15, 2.09, and 2.17, respectively. Using the asymptotic Formula (8), we obtained the following probabilities of difference between the periodic components of the seismic regime with these periods and a purely random Poisson process: 0.875, 0.882, 0.883, 0.863, and 0.886.

Thus, it can be stated that the sequence of time moments of earthquakes with magnitudes not lower than 6.5 contained a periodic component with a period of 26.64 days, close to the period of the Sun’s rotation with a probability of not less than 0.883. This fact confirms the hypothesis about the influence of solar activity on the Earth’s seismicity.

## 5. Method for Assessing the Measure of Mutual Advance of Two Streams of Events

In the future, we will be interested in the question of whether there is an advance of the moments of time of the largest local maxima of the average values of the proton flux density (Figure 2a) relative to the moments of time of earthquakes (Figure 2b). Clarification of this question requires also an assessment of the “reverse” advance and calculation of their difference. If the average value of this difference is positive, then there is a trigger effect of the proton flux on seismicity. In addition, the value of the average difference of the advance measures will give a measure of the trigger effect.

To clarify this issue, we applied the influence matrix method proposed in [40] to assess the degree of influence of earthquake sequences on each other in several seismically active regions. In its original implementation, this method is multidimensional. However, below, it is simplified and modified for the practically important situation of two time sequences. This modification was previously used in [41,42,43,44,45] to analyze the relationships between seismic event times and local extremum time points of various microseismic background statistics, magnetic field fluctuations, ground tremor, and meteorological time series properties.

Let $t_{\;j}^{\left(\alpha \right)},\;j=1,\ldots ,N_{\alpha };\;\alpha =1,2$ represent the moments of time of two sequences of events. In our case, these are

(1)a sequence of time moments corresponding to the largest local maxima of the average values of the proton flux;(2)sequence of times of seismic events with magnitude of at least 6.5.

Let us represent their intensities as follows:(9)$$\;\lambda^{\left(\alpha \right)}(t)=b_{\;0}^{\left(\alpha \right)}\;+\sum_{k=1}^{2}b_{\;k}^{\left(\alpha \right)}\cdot g^{\left(k\right)}(t)$$
where $b_{\;0}^{\left(\alpha \right)}\geq 0,b_{\beta }^{\left(\alpha \right)}\geq 0$ are parameters, $g^{\left(\beta \right)}(t)\;$—function of influence of time moment $t_{j}^{\left(\beta \right)}$ of the sequence with number β:(10)$$g^{\left(\beta \right)}(t)=\sum_{t_{j}^{\left(\beta \right)}<t}e^{-(t-t_{\;j}^{\left(\beta \right)})/\tau }$$

According to Formula (10), the weight of the event with number j becomes non-zero for times $t\;>t_{j}^{\left(\beta \right)}$ and decays with characteristic time τ. The parameter $b_{\beta }^{\left(\alpha \right)}$ determines the degree of influence of the flow β on the flow α. The parameter $b_{\alpha }^{\left(\alpha \right)}$ determines the degree of influence of the flow α on itself (self-excitation), and the parameter $b_{\;0}^{\left(\alpha \right)}$ reflects a purely random (Poisson) component of intensity. Let us fix the parameter τ and consider the problem of determining the parameters $b_{\;0}^{\left(\alpha \right)},b_{\beta }^{\left(\alpha \right)}$.

The log-likelihood function for a non-stationary Poisson process is equal to over the time interval [0,T] [38]:(11)$$\ln(L_{\alpha })=\sum_{j=1}^{N_{\alpha }}\ln(\lambda^{\left(\alpha \right)}(t_{j}^{\left(\alpha \right)}))-\int_{0}^{T}\lambda^{\left(\alpha \right)}(s)ds,\;\alpha =1,2$$

It is necessary to find the maximum of functions (11) with respect to the parameters $b_{\;0}^{\left(\alpha \right)},b_{\beta }^{\left(\alpha \right)}$. Taking into account Formula (11), we can write the derivative of the logarithmic likelihood function with respect to the parameters:(12)$$\frac{\partial \ln(L_{\alpha })}{\partial b_{\;0}^{\left(\alpha \right)}}=\sum_{j=1}^{N_{\alpha }}\frac{1}{\lambda^{\left(\alpha \right)}(t_{j}^{\left(\alpha \right)})}-\int_{0}^{T}ds,\;\frac{\partial \ln(L_{\alpha })}{\partial b_{\;\beta }^{\left(\alpha \right)}}=\sum_{j=1}^{N_{\alpha }}\frac{g^{\left(\beta \right)}(t)}{\lambda^{\left(\alpha \right)}(t_{j}^{\left(\alpha \right)})}-\int_{0}^{T}g^{\left(\beta \right)}(s)ds$$

From where and from Formula (9) it follows:(13)$$\begin{array}{l} b_{\;0}^{\left(\alpha \right)}\frac{\partial \ln(L_{\alpha })}{\partial b_{\;0}^{\left(\alpha \right)}}+\sum_{\beta =1}^{2}b_{\beta }^{\left(\alpha \right)}\frac{\partial \ln(L_{\alpha })}{\partial b_{\beta }^{\left(\alpha \right)}}=\sum_{j=1}^{N_{\alpha }}\left\{\frac{b_{\;0}^{\left(\alpha \right)}\;+\sum_{k=1}^{2}b_{\;k}^{\left(\alpha \right)}\cdot g^{\left(k\right)}(t)}{\lambda^{\left(\alpha \right)}(t_{j}^{\left(\alpha \right)})}\right\}-\\ -\int_{0}^{T}(b_{\;0}^{\left(\alpha \right)}\;+\sum_{k=1}^{2}b_{\;k}^{\left(\alpha \right)}\cdot g^{\left(k\right)}(s))ds=N_{\alpha }-\int_{0}^{T}\lambda^{\left(\alpha \right)}(s)ds \end{array}$$

Since the parameters $b_{\;0}^{\left(\alpha \right)},b_{\beta }^{\left(\alpha \right)}$ must be non-negative, each term in the leftmost part of this formula is equal to zero at the point of maximum of function (11)—either due to the necessary conditions of the extremum (if the parameters are positive), or, if the maximum is reached at the boundary, then the parameters themselves are equal to zero. Consequently, at the point of maximum of the likelihood function, the equality is satisfied:(14)$$\int_{0}^{T}\lambda^{\left(\alpha \right)}(s)ds=N_{\alpha }$$

Let us substitute the expression $g^{\left(\beta \right)}(t)$ from (9) into (14) and divide by T. Then, we get another form of Formula (14):(15)$$b_{\;0}^{\left(\alpha \right)}+\sum_{\beta =1}^{2}b_{\beta }^{\left(\alpha \right)}\cdot {\bar{g}}^{\left(\beta \right)}=\lambda_{\;0}^{\left(\alpha \right)}\equiv N_{\alpha }/T$$
where(16)$${\bar{g}}^{\left(\beta \right)}=\int_{0}^{T}g^{\left(\beta \right)}(s)ds/T$$

Substituting $b_{\;0}^{\left(\alpha \right)}$ from (15) into (11), we obtain the following maximum problem:(17)$$\Psi^{\left(\alpha \right)}(b_{1}^{\left(\alpha \right)},b_{2}^{\left(\alpha \right)})=\sum_{j=1}^{N_{\alpha }}\ln(\lambda_{\;0}^{\left(\alpha \right)}+\sum_{\beta =1}^{2}b_{\beta }^{\left(\alpha \right)}\cdot \Delta g^{\left(\beta \right)}(t_{j}^{\left(\alpha \right)}))\to \max$$

Here, $\Delta g^{\left(\beta \right)}(t)=g^{\left(\beta \right)}(t)-{\bar{g}}^{\left(\beta \right)}$, under restrictions:(18)$$b_{1}^{\left(\alpha \right)}\geq 0,b_{2}^{\left(\alpha \right)}\geq 0,\sum_{\beta =1}^{2}b_{\beta }^{\left(\alpha \right)}{\bar{g}}^{\left(\beta \right)}\leq \lambda_{\;0}^{\left(\alpha \right)}$$

Function (17) is convex with negative definite Hessian [40] and, therefore, problem (17)–(18) has a unique solution. Having solved problem (17)–(18) numerically for a given τ, we can introduce the elements of the influence matrix $\kappa_{\beta }^{\left(\alpha \right)},\alpha =1,2;\;\beta =0,1,2$ according to the formulas:(19)$$\kappa_{0}^{\left(\alpha \right)}=b_{\;0}^{\left(\alpha \right)}/\lambda_{\;0}^{\left(\alpha \right)}\geq 0,\;\kappa_{\beta }^{\left(\alpha \right)}=b_{\beta }^{\left(\alpha \right)}\cdot {\bar{g}}^{\left(\beta \right)}/\lambda_{\;0}^{\left(\alpha \right)}\geq 0$$

The quantity $\kappa_{0}^{\left(\alpha \right)}$ is a share of the average intensity $\lambda_{\;0}^{\left(\alpha \right)}$ of the process with number α, which is purely stochastic, the part $\kappa_{\alpha }^{\left(\alpha \right)}$ is caused by the influence of self-excitation α→α and $\kappa_{\beta }^{\left(\alpha \right)},\;\beta \neq \alpha$ is determined by the external influence β→α. From Formula (15) follows the normalization condition:(20)$$\kappa_{0}^{\left(\alpha \right)}+\sum_{\beta =1}^{2}\kappa_{\beta }^{\left(\alpha \right)}=1,\;\alpha =1,2$$

As a result, we can determine the influence matrix:(21)$$\left(\begin{array}{l} \kappa_{\;\;0}^{\left(1\right)}\;\kappa_{\;\;1}^{\left(1\right)}\;\kappa_{\;\;2}^{\left(1\right)}\\ \kappa_{\;\;0}^{\left(2\right)}\;\kappa_{\;1}^{\left(2\right)}\;\kappa_{\;2}^{\left(2\right)} \end{array}\right)$$

The first column of matrix (21) is composed of Poisson shares of mean intensities. The diagonal elements of the right submatrix of size 2 × 2 consist of self-excited elements of mean intensity, while the off-diagonal elements correspond to mutual excitation. The sums of the component rows of the influence matrix (21) are equal to 1. The influence matrices are estimated in a certain sliding time window of length with offset and with a given value of the attenuation parameter τ.

When analyzing variations of the components of influence matrices in sliding time windows corresponding to the mutual influence of the analyzed time sequences, the main attention is paid to their local maxima with their subsequent averaging. Let $M_{L}$ be the number of windows lengths within limits from $L_{\min}$ up to $L_{\max}$. Thus, the sequence of windows lengths is $L_{j}=L_{\min}+(j-1)\Delta L,\;j=1,\ldots ,M_{L}$, where $\Delta L=(L_{\max}-L_{\min})/(M_{L}-1)$. Each time window of the length $L_{j}$ is shifted along time axis with mutual shift Δt. Let $t_{k}(L_{j})$ be the sequence of time moments corresponding to right ends of time windows of the length $L_{j}$. The number $K(L_{j})$ of time moments $t_{k}(L_{j})$ is defined by mutual shift Δt of time windows of the length $L_{j}$. Let $\left(t_{k}(L_{j}),\;c_{\;k}^{\left(1\right)}(L_{j})\right)$ and $\left(t_{k}(L_{j}),\;c_{\;k}^{\left(2\right)}(L_{j})\right)$ be elements $\kappa_{\;\;2}^{\left(1\right)}$ and $\kappa_{\;1}^{\left(2\right)}$ of the matrix (21), corresponding to mutual influences 2→1 and 1→2 of analyzed time moments for current position $t_{k}(L_{j})$ of time window of the length $L_{j}$. Let $\left(t_{k}^{*}(L_{j}),\;{\hat{c}}_{\;k}^{\left(\alpha \right)}(L_{j})\right)$, α=1,2 be local maxima of $c_{\;k}^{\left(\alpha \right)}(L_{j})$, i.e., $c_{\;k-1}^{\;(\alpha )}(L_{j}))<{\hat{c}}_{\;k}^{\left(\alpha \right)}(L_{j})<c_{\;k+1}^{\;(\alpha )}(L_{j})$.

Let us take some “small” time interval of the length η and for the sequence of time moments $\left[\nu_{m},\nu_{m+1}\right]$, $\nu_{m+1}-\nu_{m}=\eta$ of such time fragments, where we will calculate the mean values $G_{2\to 1}(\nu_{m+1})$ and $G_{1\to 2}(\nu_{m+1})$ of ${\hat{c}}_{\;k}^{\left(\alpha \right)}(L_{j})$ for which their time marks $t_{k}^{*}(L_{j})$ belong to these fragments. Averaging is performed over all time window lengths $L_{j},\;j=1,\ldots ,M_{L}$. These mean values in dependence on the right end of intervals $\nu_{m+1}$ gives the measures of the averaged effects of the advance of second sequence of time moments with respect to the first one and vice versa. Our main purpose is calculating the difference $\Delta G(\nu_{m+1})=G_{2\to 1}(\nu_{m+1})-G_{1\to 2}(\nu_{m+1})$. In this formula, the first sequence is the sequence of time moments of earthquakes with a magnitude not less than 6.5, whereas the second sequence is time moments of the largest local maxima of the mean proton flux time series. Thus, if average $<\Delta G(\nu_{m+1})>$ is positive, it means that there is a trigger effect.

The full set of parameters of the method is the following: τ, $L_{\min}$, $L_{\max}$, $M_{L}$, Δt, η. In our calculations, we used τ=0.05 year (approximately 18 days), $L_{\min}=0.5$ year, $L_{\max}=1$ year, $M_{L}=100$, Δt=1 day, η=0.1 year. The calculation results are most sensitive to the choice of parameters τ, $L_{\min}$, $L_{\max}$. The values used were chosen as a result of trial calculations and selection of the best options.

## 6. Measures of Mutual Advance of Local Maxima of the Average Value of Proton Flux Density and Seismic Event Sequence

In this section of the paper, we apply the method described earlier to the analysis of the relationships between the time sequences presented in Figure 2. Figure 5 presents the results of such an analysis.

From the comparison of the graphs in Figure 5, it is evident that the advance of the moments of time of the largest local maxima of the smoothed proton flux relative to the moments of time of earthquakes with a magnitude of not less than 6.5 was on average significantly greater than the reverse advance. At the same time, the difference between the average values of the advance measures, presented in Figure 5c, had a positive average value equal to 0.17. This value can be interpreted as an estimate of the part of the average intensity of all earthquakes with a magnitude of not less than 6.5 for those seismic events for which the maximum values of the average proton flux density are a trigger. Another numerical characteristic of the trigger effect is the part of the lengths of the time interval for which the difference between the “direct” and “reverse” advance is positive. For the graph in Figure 5c, this part is equal to 0.62.

## 7. Proton Flux Density Time Series Statistics

The average proton flux density values used above are the simplest statistics. An idea arises to try other proton time series statistics and to estimate the relationship of the times of their “most expressive” (i.e., largest local maxima or smallest local minima) extreme values with the times of earthquakes using the above model of influence matrices. In addition to the simple average values, we used five different proton flux time series statistics described below. These statistics were estimated in the same time windows of 1440 5 min samples (5 days), taken with an offset of 288 samples (1 day), as before, when calculating the average values.

(1) *The kurtosis* of a time series x(t) is calculated in each time window using the following formula [45]:(22)$$\kappa =<{\left(x(t)-m_{x}\right)}^{4}>/{\left(<{\left(x(t)-m_{x}\right)}^{2}>\right)}^{2},\;m_{x}=<x(t)>\;$$

Here, the angle brackets denote the operation of calculating the mean value. The value κ can be considered as a measure of the difference from the Gaussian distribution, for which κ=3. Below, we use the logarithm of the kurtosis coefficient: lg(κ).

(2) *The minimum wavelet-based normalized entropy* En of a time series x(t) is calculated based on the decomposition of the time series within a window into orthogonal wavelets.(23)$$En=-\sum_{k}p_{k}\cdot \log(p_{k})/\log(N)$$

In Formula (23), $p_{k}=c_{k}^{2}/\sum_{j}c_{j}^{2}$, $c_{k}$ are the wavelet coefficients of the signal x(t), and N is the total number of wavelet coefficients. Seventeen orthogonal Daubechies wavelets were used: 10 ordinary bases with a minimum support with a number of vanishing moments from 1 to 10 and 7 so-called Daubechies symlets [46], with a number of vanishing moments from 4 to 10. For each of the bases, the entropy (23) of the distribution of the squares of the wavelet coefficients was calculated, and then, by enumeration, the optimal basis was found that realized the minimum value in each time window. By construction, 0≤En≤1. The details of calculating the entropy (23) in a sliding time window are described in [47].

(3) *Wavelet-based spectral slope*
β. After determining the optimal orthogonal wavelet from the minimum entropy condition, it is possible to calculate the average values $S_{k}$ of the squares of the wavelet coefficients at each detail level, which is part of the oscillation energy corresponding to the detail level with the number k, which corresponds to the frequency band with the boundary frequencies $f_{\min}^{\left(k\right)}=1/(2^{\left(k+1\right)}\Delta s)$ and $f_{\max}^{\left(k\right)}=1/(2^{k}\Delta s)$, where Δs is the length of the sampling time interval (in our case Δs = 5 min) [46]. Let us consider the values of the periods corresponding to the centers of these frequency bands:(24)$$T_{k}=2/(f_{\min}^{\left(k\right)}+f_{\max}^{\left(k\right)})\;=\;2\Delta s/(2^{-k}+2^{-(k+1)})$$

The quantities $S_{k}=S(T_{k})$ are similar to the Fourier power spectra. These quantities are convenient to use when calculating the slope of the graph of the logarithm of the power spectrum as a function of the logarithm of the period. The spectral slope in each time window is found by the least squares method:(25)$$\sum_{k}(\ln(S(T_{k}))-\beta \cdot \ln(T_{k}))-c{)}^{2}\to \underset{\beta ,c}{\min}$$

(4) *The Donoho–Johnston wavelet-based index (DJ-index*) γ is defined as the ratio of the number of “large” wavelet coefficients by absolute value to their total number. By definition, 0≤γ≤1. The threshold separating the “large” wavelet coefficients is $T_{DJ}=\sigma \sqrt{2\cdot \ln N}$. This threshold separates the informative wavelet coefficients from other coefficients that are considered noisy [46,48]. The value σ is an estimate of the standard deviation of noise under the assumption that the noise is most concentrated at the first detail level of the orthogonal wavelet decomposition. To estimate the value, the median estimate of the standard deviation of a normal random variable is used:(26)$$\sigma =med\left\{\left|c_{k}^{\left(1\right)}\right|\;,\;k=1,\ldots ,N/2\right\}/0.6745$$

(5) *The multifractal singularity spectrum support width*
Δα is an important characteristic of the signal and is considered as a measure of the diversity (complexity) of its stochastic behavior. It is defined as $\Delta \alpha =\alpha_{\max}-\alpha_{\min}$, where $\alpha_{\min}$ and $\alpha_{\max}$ are estimates of minimum and maximum values of the Holder–Lipschitz exponent [49] α, which governs the behavior of the signal at the vicinity of time moment t: $|x(t+\delta /2)-x(t-\delta /2)|\;∼\;|\delta {|}^{\;\alpha },\;\delta \to 0$. For a mono-fractal signal, the Holder–Lipschitz exponent is the same for all time moments t. Otherwise, the signal is multi-fractal, and the concept of the spectrum of singularities F(α) is introduced, equal to the fractal dimension of the time moments with the same value of the Holder–Lipschitz exponent, equal to α [50]. To estimate Δα in each time window, we used the method of fluctuation analysis after removing scale-dependent trends [51]. The implementation of the method used is described in detail in [47]. To remove local polynomial trends for the proton flux density time series, we used zero-order polynomials, i.e., we analyzed fluctuations after removing local means.

For a sequence of time intervals of 5 days, taken with a shift of 1 day, we calculated the values of all five statistics of the proton flux density time series. The results of these calculations are presented in the graphs in Figure 6.

From the graphs of the power spectra of the time series of changes in statistics, it is evident that for all of them, with the exception of lg(κ), there is a periodicity of 89 days, which is especially pronounced for γ and Δα. It should be noted that the power spectrum of the change in the average values of the proton flux density does not contain a spectral component with a period of 89 days.

Let us consider in more detail the time–frequency structure of the variations in the singularity spectrum support width Δα (Figure 6(a5)), for which the 89-day periodicity is most clearly visible. Let us denote Δα(s) the dependence of the singularity spectrum carrier width on time (the position of the right end of the 5-day time window with a 1-day offset) s and calculate the Morlet wavelet transform [46]:(27)$$c_{\Delta \alpha }(t,\upsilon )=\frac{1}{\sqrt{\upsilon }}\int_{-\infty }^{+\infty }\Delta \alpha (s)\cdot \varphi \left(\frac{s-t}{\upsilon }\right)ds,\;\upsilon >0,\;\varphi (t)=\frac{1}{\pi^{\;1/4}}\exp(-t^{2}/2-i\pi t)$$

The values $|c_{\Delta \alpha }(t,\upsilon ){|}^{2}$ can be interpreted as the energy of signal Δα(s) oscillations in the vicinity of a time point t with a period υ. Figure 7a shows the Morlet time-frequency diagram of values $\mathrm{lg}|c_{\Delta \alpha }(t,\upsilon )|$ for 200 values of periods υ varying within the range from 10 to 500 days with a uniform step on a logarithmic scale. For the frequency band with periods from 63 to 158 days (logarithms of periods from 1.8 to 2.2), in which the most intense periodic variations of Δα(s) with a central period of 89 days are concentrated, we calculated the maximum values $\underset{\upsilon \;}{\max}\left(\mathrm{lg}|c_{\Delta \alpha }(t,\upsilon )|\right)$. These maximum values are shown in Figure 7b by a black line. The red line in Figure 7b shows the cyclic trend for the maximum values of the logarithms of the Morlet wavelet coefficients in the frequency band highlighted above. The period of this oscillation was determined numerically from the condition of minimum variance of deviations for trial cyclic trends with periods in the range from 1500 to 5500 days. As a result of such calculations, it turned out that the optimal period is equal to 4429 days or approximately 12.13 years, that is, very close to the period of solar cycles.

## 8. Measures of Mutual Advance of Local Extrema of Proton Flux Density and Seismic Event Sequence Statistics

The further plan of using five statistics of the proton flux density time series consists of assessing the measures of advancement of the time moments of their most expressive local extrema (the largest local maxima and the smallest local minima) relative to the time moments of earthquakes with a magnitude of at least 6.5. In this case, the number of points of the most expressive local extrema will be chosen equal to the number of seismic events, i.e., 1136.

To eliminate the influence of low-frequency components of the change in the values of statistics on the determination of the moments of time of local extremes, the time series, the graphs of which are presented in Figure 6(a1–a5), were subjected to the operation of removing low frequencies using Gaussian kernel smoothing. Let u(t) be a time series with discrete time t. Gaussian kernel averaging of a time series u(t) with radius (scale parameter) h>0 at the moment of time t, is calculated using the following formula [52]:(28)$$\bar{u}(t|h)=\sum_{s}u(s)\cdot e^{-{\left(\frac{t-s}{h}\right)}^{2}}/\sum_{s}e^{-{\left(\frac{t-s}{h}\right)}^{2}}$$

Calculation of the kernel averaging by Formula (28) for long time series can be effectively implemented using the fast Fourier transform. Then, the average values of the time series for the averaging radius h equal to 2 days were subtracted from the time series of changes in statistics and the most expressive points of local extrema were found for the residuals. These operations are illustrated by the graphs in Figure 8.

When estimating the advance measures by the points of local extrema of the proton flux statistics, we tested both the points of the largest local maxima and the points of the smallest local minima after excluding low frequencies using Gaussian smoothing (28). In this case, the differences between the “direct” and “reverse” lead were calculated. Then, the variant of the largest local maxima and the smallest local minima for which the average value of the difference between the average measures of the “direct” and “reverse” lead was maximum was selected. As a result of such an enumeration of variants, it turned out that the most preferable were the smallest points of local minima for the statistics lg(κ), β, γ, and Δα, whereas the largest local maxima for the entropy was En. The results of estimating the differences in the lead measures are presented in Figure 9.

It is interesting to note that when analyzing the prognostic properties of low-frequency seismic noise measured by a global network of 229 broadband seismic stations located around the world, it turned out that it is the points of the smallest local minima of statistics γ, Δα, and the points of the largest local maxima of entropy En that have the maximum prognostic effects relative to the times of the strongest earthquakes with magnitudes of at least 7 [41].

Another characteristic of the difference of average lead measures is the part of interval lengths with positive values of the difference. This characteristic is equal to $\mathrm{lg}{\left(\kappa \right)}_{\min}$—0.78; $\beta_{\min}$—0.75; $\gamma_{\min}$—0.71; $\Delta \alpha_{\min}$—0.83; $En_{\max}$—0.76; Mean—0.95, that is, the use of averaging provides a frequent positive value of the lead measure, although it loses out in comparison with the average value of using minima Δα.

## 9. Discussion

The conclusions of the article were obtained as a result of applying a sequence of methods. The first stage consisted of a simple check for the presence of a 27-day period in the seismic event sequence, which dominates the proton flux density time series and is related to the rotation of the Sun. At this stage, the existence of this periodicity in the seismic event stream was confirmed with a probability of 88%. This is an indirect confirmation of the influence of the proton flux on seismicity. At the second stage of the analysis, the hypothesis about the influence of the proton flux on earthquakes was tested by a more “direct” method, based on the influence matrix method. In this method, two time sequences are processed, and the influence of events in each sequence to events in the other stream is directly estimated. In this case, the method allows for the calculating the contribution of a purely random (Poisson) component, the contribution of self-excitation, and the contributions of mutual excitation. In this analysis, one of the event sequences is always a sequence of time moments of earthquakes with a magnitude of at least 6.5. As for the second sequence of times, it varies. The simplest option is to select the time points of the largest local maxima of the average value of the proton flux density. For this option, the proportion of the average intensity of earthquakes for which the maximum values of the proton flux density are a trigger is 17% (Figure 5). However, this result can be significantly improved if, instead of a simple average, we take more sophisticated statistics of the behavior of the time series of the proton flux density. The results for enumerating five variants of statistics are shown in Figure 9. The best result (28%) was obtained when using the time points of the smallest local minima of the multifractal singularity spectrum support width (Figure 9d). But, if we average over the set of all used statistics, we obtain 23% (Figure 9f). Although the second value is less than the first, when averaging, the proportion of time when the trigger effect occurs is 0.95, while for the “record” statistics it is 0.83. In this sense, 23% is a more stable estimate.

An 89-day periodicity in the variations of proton flux statistics has been revealed. One hypothesis is that this periodicity may be related to the modulation of the proton flux density by the motion of Mercury, the planet closest to the Sun with an orbital period of 89 days. The presence of a 12-year periodicity in the change in the maximum values of the logarithms of the modules of the Morlet wavelet coefficients for the singularity spectrum support width confirms the connection of the 89-day periodicity with solar dynamics. Another hypothesis for the origin of this periodicity is the coincidence of 89 days with half the oscillation period of the SOHO satellite, which measures the proton flux density, in the vicinity of the Lagrange libration point L1 [32]. However, the mechanism of such modulation, which is maximum precisely for the Donoho–Johnston index statistics and the singularity spectrum support width, remains unclear.

## Figures and Tables

**Figure 1 entropy-27-00505-f001:**
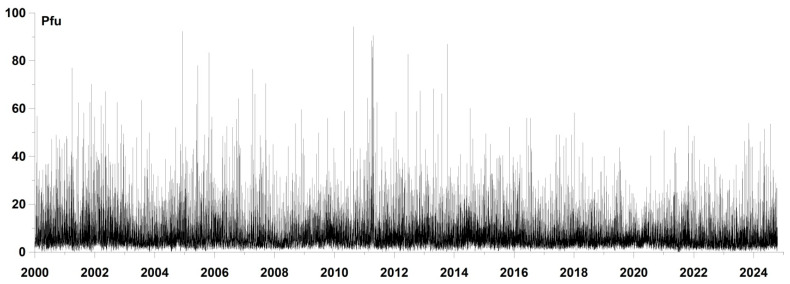
Time series graph of the proton flux from the beginning of 2000 to 17 October 2024 with a time step of 5 min. Proton flux density unit “Pfu” means Particles⋅cm^−1^·s^−1^·steradian^−1^.

**Figure 2 entropy-27-00505-f002:**
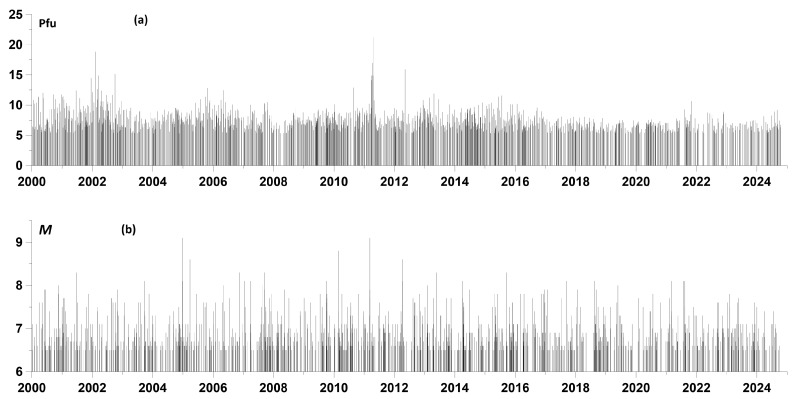
(**a**) Time sequence of 1136 largest local maxima of average values of proton flux density in sliding time windows of 5 days with a shift of 1 day; (**b**) time sequence of earthquakes with a magnitude of at least 6.5; data from the source [34].

**Figure 3 entropy-27-00505-f003:**
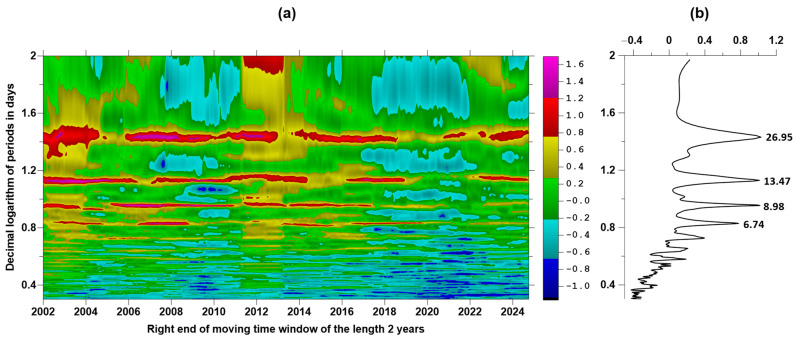
Spectral composition of the proton flux in the low-frequency part of the spectrum: (**a**) time–frequency diagram of the evolution of the logarithm of the power spectrum in a sliding time window of 2 years; (**b**) graph of the average power spectrum from all time windows, where the period in days is shown next to the 4 largest spectral peaks.

**Figure 4 entropy-27-00505-f004:**
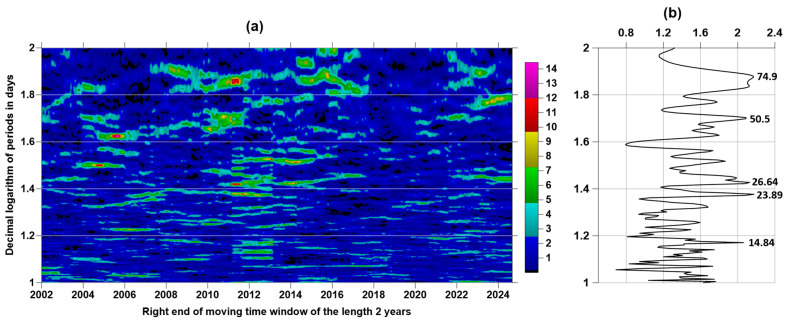
(**a**) Time–frequency diagram of the evolution of the increment of the logarithmic likelihood function in a sliding time window of 2 years; (**b**) graph of the average values of the increment of the logarithmic likelihood function from all time windows, where the period in days is shown next to the 5 largest spectral peaks.

**Figure 5 entropy-27-00505-f005:**
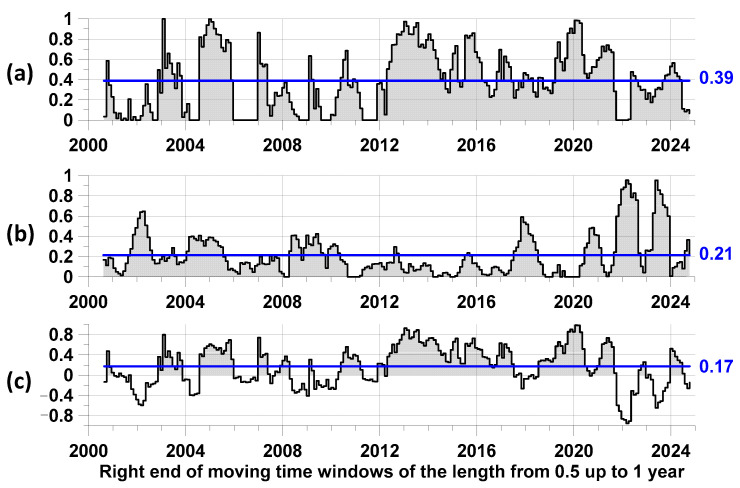
(**a**) Average values of local maxima of the component of influence matrices corresponding to the “direct” influence of the moments of time of the largest local maxima of the averaged values of the proton flux density on the moments of time of earthquakes with a magnitude of at least 6.5; (**b**) average values of local maxima of the component of influence matrices corresponding to the “reverse” influence of the moments of time of earthquakes on the moments of time of the largest local maxima of the averaged values of the proton flux density; (**c**) the difference between the average values of local maxima of the component of influence matrices corresponding to the “direct” influence and corresponding to the “reverse” influence. Blue line—average value; their numerical values are indicated on the right.

**Figure 6 entropy-27-00505-f006:**
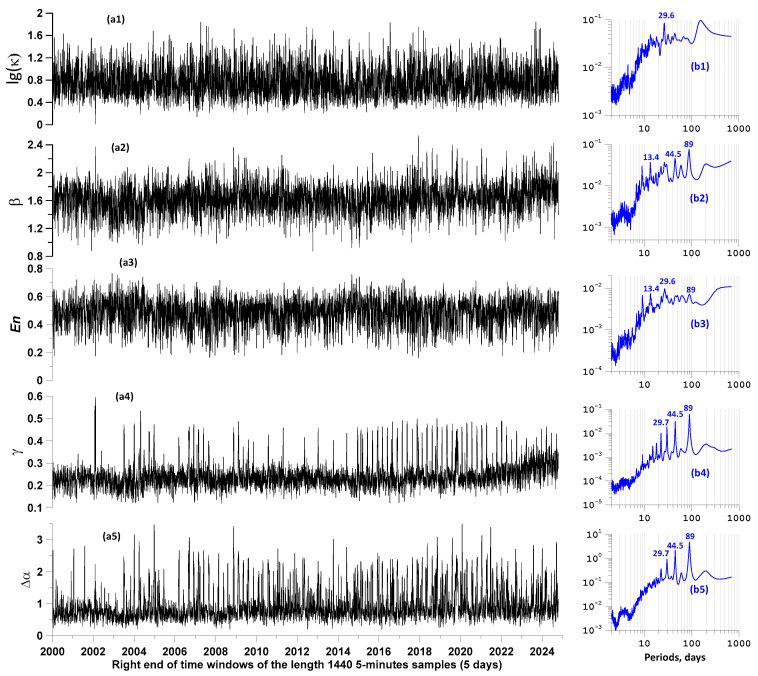
(**a1**–**a5**) Graphs of time series of 5 proton flux density statistics: logarithm of kurtosis lg(κ), wavelet-based spectral slope β, minimum normalized entropy of wavelet coefficients En, Donoho–Johnstone index γ, and the singularity spectrum support width Δα, calculated in sliding time windows of 5 days with a shift of 1 day. On the right are graphs (**b1**–**b5**), corresponding to the power spectra of time series of proton flux density statistics; for the largest spectral peaks, their periods in days are indicated.

**Figure 7 entropy-27-00505-f007:**
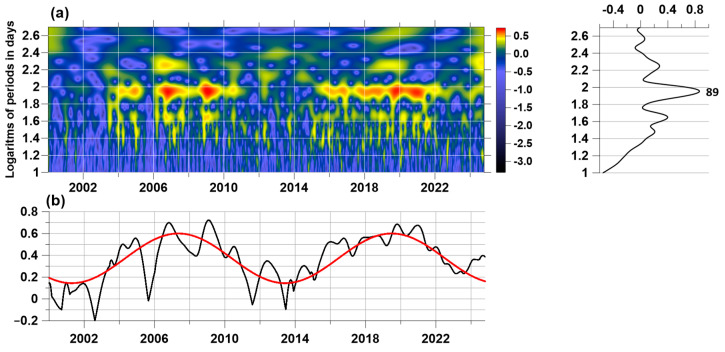
(**a**) Time–frequency diagram of the logarithms of the moduli of the Morlet wavelet coefficients of the variations of the singularity spectrum support width Δα for periods from 10 to 500 days; (**b**) the black line represents the maximum values of the logarithms of the moduli of the Morlet coefficients for periods from 63 to 158 days, where the red line is the optimal cyclic trend with a period of 4429 days or 12.13 years.

**Figure 8 entropy-27-00505-f008:**
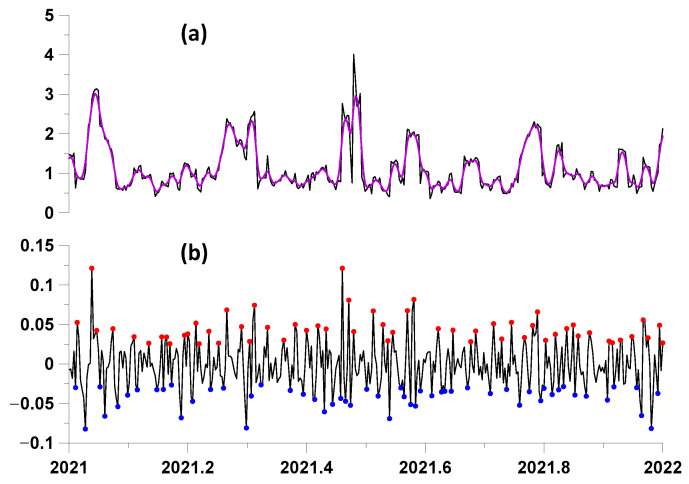
(**a**) The black line shows a graph of a fragment of the time series of changes in singularity spectrum support width Δα for the time interval 2021–2022; the purple line is the smoothing of the time series by a Gaussian kernel of radius 2 days. In (**b**), the red and blue dots show the positions of 1136 largest local maxima and minima of the difference between the original changes in statistics Δα and the smoothed values (black line) in this time fragment.

**Figure 9 entropy-27-00505-f009:**
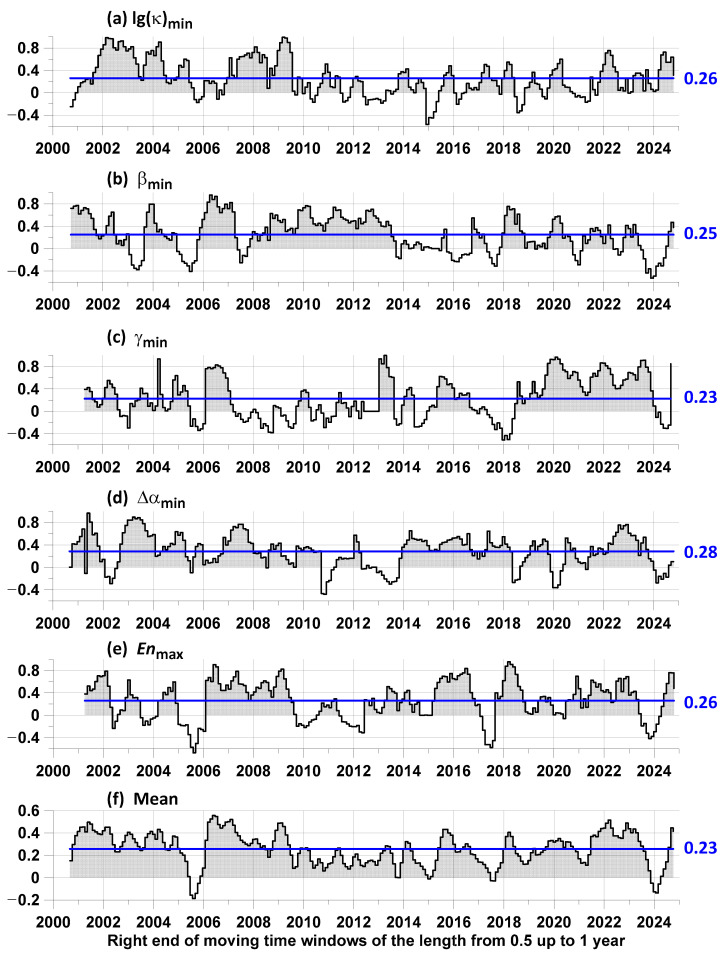
(**a**–**e**) Graphs of differences between average values of local extrema of components of influence matrices corresponding to “direct” advance of time points of local extrema of 5 proton flux density statistics relative to time moments of earthquakes with magnitude not lower than 6.5 and corresponding to “reverse” determination. Graphs (**a**–**d**) were constructed, respectively, for local minima of logarithm of kurtosis lg(κ), spectral slope β, Donoho–Johnstone index γ, and singularity spectrum support width Δα; graph (**e**) shows points of local maxima of entropy En. Graph (**f**) is the averaging of curves in graphs (**a**–**e**). Blue lines are average values; their numerical values are indicated on the right. From the point of view of the average value of the difference in the advance measures, all these results are noticeably better than using the simplest statistics—the average value of the proton flux density (Figure 5c). In this case, the best result is achieved when using the minimum Δα values—the multi-fractal singularity spectrum support width, for which the average value is 0.28.

## Data Availability

The proton flux data were available from the site https://soho.nascom.nasa.gov/data/data.html, accessed on 1 January 2025. The data of earthquakes sequence are available from the site https://earthquake.usgs.gov/earthquakes/search/, accessed on 2 February 2025.

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
