# Peer review of "Quantitative Assessment of the Trigger Effect of Proton Flux on Seismicity"

_entropy, 2025, doi:10.3390/e27050505_

Round 1
Reviewer 1 Report
Comments and Suggestions for Authors
I read carefully the manuscript. In this work, the authors have studied the relationship between the influence of the proton flux density coming from the Sun on the sequence of earthquakes with
a magnitude of at least 6.5.
In my opinion the manuscript is interesting because the analysis is focused to estimate the correlation between the proton flux coming from the Sun with the sequence of earthquakes. However, there are a few of points that authors must to take into account.
1) Here, authors did not specify the seismic region where the sequence of earthquakes were monitored. They have to cite the data set where the seismic catalogue can be visited.
2) Authors must to clarify why the Gutenberg-Richter parameters are not important.
3) In section 4, devoted to estimate the periodicity of the earthquakes sequence, In order to keep the reading the main text, I suggest the procedure from line 136 to 176 could be written in an appendix.
4) The same suggestion in the last point, for the procedure described from 216 to 274.
5) Line 451: ...by formula (32) for long time series... Formula (32) does not exist.
I think the manuscript can be published with the previous minor corrections.
Author Response
1) Here, authors did not specify the seismic region where the sequence of earthquakes were monitored. They have to cite the data set where the seismic catalogue can be visited.
The analysis was performed for seismic events with magnitudes not less than 6.5 all over the world. The information about seismicity was taken from the site of USGS – it has the number [34] in the reference list. Within text it was written in the Figure 2 caption. The analysis could be (and should be) performed for a number of regions, for instance for regions around Pacific Fire Ring. But this will increase essentially the volume of the paper and should be the purpose of further investigations.
2) Authors must to clarify why the Gutenberg-Richter parameters are not important.
I suppose that by Gutenberg-Richter parameters the b-value is understood. But it does not really included into analysis just because it was not necessary. It is important that the minimum magnitude of 6.5 is representative for the entire world for the time interval under consideration since the beginning of 2000. That is, all events with a magnitude of 6.5 and higher fall on the straight-line section of the recurrence law. This is indeed the case. I have inserted the following sentence into the new version of the paper after the Figure 2: “The chosen minimum magnitude of 6.5 is representative for the whole world and the corresponding time sequence does not contain aftershocks, which is important to ensure homogeneity of the time points of seismic events.”
3) In section 4, devoted to estimate the periodicity of the earthquakes sequence, In order to keep the reading the main text, I suggest the procedure from line 136 to 176 could be written in an appendix.
4) The same suggestion in the last point, for the procedure described from 216 to 274.
It seems to me that placing the exposition of mathematical methods of data analysis in the appendix will make it difficult to read the text, since the potential reader will be forced to constantly look at the end of the article to understand the content. In addition, the specificity of the Entropy journal lies in its mathematical focus. Therefore, I will allow myself to disagree with this proposal of the reviewer and leave the exposition of the methods used in its current form.
5) Line 451: ...by formula (32) for long time series... Formula (32) does not exist.
Thank you. It was a typo. It should actually be (29). It's all fixed now.
Reviewer 2 Report
Comments and Suggestions for Authors
The authors have proposed applying several methods to compare the solar proton flux with the seismic activity, using a data set of more than twenty years. There are several analyses and several results. However, it is hard to extract knowledge of such results for the reader who is not used to such methods.
My decision is that some revisions are needed:
1) Check the self-citations and double-check if they are necessary.
2) Some lines to explain more about the methods used to compare and analyze different data sets. Also, it is not clear from the text how these different methods of analysis complement each other, i.e., their advantages and disadvantages.
3) There are several analyses; however, the discussion is short and does not go deep into the consequences or comparison between the results. The discussion does not make clear the paper's new contributions to the respective research area. In the present form of the discussion, it seems the reader must try to find the significance of the results and the paper.
Dear Editor,
There are nine self-citations in the text. However, they are related to the field of the work. The authors could cite other sources, but these self-references do not seem problematic.
Author Response
Responses to Reviewer #2
1) Check the self-citations and double-check if they are necessary.
In the file of open review I have found the next sentences:
“Dear Editor,
There are nine self-citations in the text. However, they are related to the field of the work. The authors could cite other sources, but these self-references do not seem problematic.”
Thus, I suppose that this is a response to this comment.
2) Some lines to explain more about the methods used to compare and analyze different data sets. Also, it is not clear from the text how these different methods of analysis complement each other, i.e., their advantages and disadvantages.
3) There are several analyses; however, the discussion is short and does not go deep into the consequences or comparison between the results. The discussion does not make clear the paper's new contributions to the respective research area. In the present form of the discussion, it seems the reader must try to find the significance of the results and the paper.
I have essentially increased the Discussion section – please look the new variant of the text. I hope that I explained all steps of the used method.
Round 2
Reviewer 1 Report
Comments and Suggestions for Authors
This final version has been improved, In my opinion the manuscript can be published.